# Evidence of ascariasis in a Celtic newborn from northern Italy

**Ramón López-Gijón**[1,2,10], **Wolf-Rüdiger Teegen**[3,4/+], **Zita Laffranchi**[5,6],
**Daniele Vitali**[7], **Albert Zink**[8], **Marco Milella**[5,9]

[1]Universidad de Granada, Facultad de Medicina, Laboratorio de Antropología, Granada, Spain
[2]Universidade de Évora, Laboratório Hercules, Évora, Portugal
[3]Ludwig-Maximilians-Universität München, Institut für Vor- und Frühgeschichtliche Archäologie und Provinzialrömische Archäologie, München, Germany
[4]Ludwig-Maximilians-Universität München, ArchaeoBioCenter, München, Germany
[5]Universität Bern, Institut für Rechtsmedizin, Abt Anthropologie, Bern, Switzerland
[6]Bernisches Historisches Museum, Bern, Switzerland
[7]Université de La Bourgogne, Dijon, France
[8]Institute for Mummy Studies, Eurac Research, Bolzano, Italy
[9]Università di Pisa, Dipartimento di Biologia, Pisa, Italy
[10]University of Coimbra, Research Centre for Anthropology and Health, Department of Life Sciences, Coimbra, Portugal

**BACKGROUND** Infections with *Ascaris lumbricoides* can be traced back to the late Pleistocene by palaeoparasitological analysis. Even today, *Ascaris* infections are still very common worldwide.

**OBJECTIVES** In a pilot study, soil samples from the pelvic area of ten individuals from the Celtic necropolis of Povegliano Veronese (northern Italy) were examined using palaeoparasitological methods. The burials date from the 3rd to 1st century Before the Common Era (BCE).

**METHODS** The palaeoparasitological methods already proven in earlier studies were applied.

**FINDINGS** Positive evidence of *Ascaris* eggs was obtained in three individuals, including a newborn. This neonate is the focus of the article. The causes of a possible *Ascaris* infection in a newborn are discussed.

**MAIN CONCLUSIONS** It may represent the oldest documented instance of ascariasis in a neonatal individual.

Key words: *Ascaris lumbricoides* - Iron Age - Italy - neonate - palaeoparasitology

The roundworm *Ascaris lumbricoides* has a long history of infecting humans, with evidence dating back to the Pleistocene (BP), at least 25,000 years BP.[1] Despite significant medical advancements, the prevalence of this infection remains alarmingly high in many regions of the world, particularly in the Global South. Moreover, due to increasing global mobility and migration, the once-forgotten disease ascariasis is re-emerging in Europe and North America.[2]

It is estimated that approximately 25% of the world's population is affected by soil-transmitted helminths (STH),[3] with nearly one billion individuals infected by *A. lumbricoides* alone.[4] Alongside dental caries, ascariasis is considered one of the most prevalent infectious diseases globally.

Soil-transmitted helminths include not only the roundworm *A. lumbricoides* but also the whipworm *Trichuris trichiura* and the hookworms *Ancylostoma duodenale* and *Necator americanus*.[3,4]

## SUBJECTS AND METHODS

The late Celtic Gallo Roman necropolis at Località Ortaia in Povegliano Veronese (northern Italy) was excavated between 2007 and 2009 by an international team from Italy, Hungary and Germany. The excavation revealed a total of 112 inhumations and 36 cremations, dated to the 3rd to early 1st century Before the Common Era (BCE).[5] During the cleaning process of the bones, soil samples were routinely collected by WRT from the visceral surfaces of the iliac and sacral bones for further analysis and from the skull area for controls. The skeletal remains were subsequently studied by WRT from anthropological and palaeopathological perspectives.[6]

In 2022, samples from all inhumations were analysed for stable isotopes. Additionally, the pars petrosa of the temporal bone was sampled from a subset of individuals, selected based on their funerary and/or anthropological features, for palaeogenetic analysis.

Financial support: This study was supported by a Swiss National Science Foundation Grant to MM (Grant Number: 10531FL_197103/1) and by a grant from the Autonomous Province of Bolzano-Alto Adige — Department of Innovation, Research, University and Museums (Funding Decree n.9/2021) to AZ. The anthropological and palaeopathological investigations by WRT were supported in part by Universität Leipzig and Ludwig-Maximilians-Universität München.
RL-G and W-RT contributed equally to this work.
+ Corresponding author: w.teegen@lmu.de
 https://orcid.org/0000-0002-0157-2858

A pilot palaeoparasitological investigation was conducted by RLG on soil samples taken from the pelvic cavities of ten individuals from Povegliano Veronese-Ortaia. Previous studies have demonstrated that parasitic worm eggs can be preserved in soil from the intestinal areas of humans and animals.[7,8] The rehydration, homogenisation, and micro sieving (RHM) technique[9] was utilised for the palaeoparasitological study. Bright field transmitted light optical microscopy was used to visualise the samples under 100×, 400×, and 600× magnification, capturing photographs at 600× magnification.[10,11]

## RESULTS

Burial 154 contains the well-preserved remains of a neonatal individual, aged 0-3 months based on long bone measurements and dental development. Although sex estimation in immature remains is inherently challenging, the mandibular morphology suggests a female individual.[12]

Pathological examination revealed several alterations: new bone formation above the vessel impressions on both parietal bones, slight new bone formation around foramina in the right orbit (the left orbit was not preserved), and extended layers of new bone formation on the long bones. These morphological changes are consistent with a probable diagnosis of scurvy (Morbus Möller-Barlow), indicating an active disease process at the time of death. Palaeoparasitological analysis identified *A. lumbricoides* eggs in three out of ten soil samples analysed, including those from burial 154. In this individual, a total of seven decorticated *A. lumbricoides* eggs were recovered (Figs 1-2). A control sample from the skull area remained negative. The shape and size of the eggs (ovoid/elliptic and ranging between 52-63 μm in length and between 36.1-49.9 μm in width) are compatible with *A. lumbricoides*. As mentioned, the decortication of the outer layer and loss of the characteristic mamillated coat can be ascribed to taphonomic processes typical of ancient specimens. The fact that all recovered *Ascaris* eggs were decorticated suggests the influence of taphonomic processes which are likely also responsible for the small number of identified eggs in the samples. In the neighbouring Lombard burial ground of Povegliano Veronese, no parasitic eggs were detected in the 14 soil samples examined.[13]

## DISCUSSION

*Ascaris lumbricoides* requires approximately 9-11 weeks to develop from infection (via the ingestion of fertilised eggs) to sexually mature individuals.[4] Case studies have demonstrated that human neonates can be infected with *A. lumbricoides*.[14,15]

One such case, published by Chu et al., involved a two-day-old infant delivered via Caesarean section.[14] A living female worm was recovered from the infant's faeces, alongside fertilised *Ascaris* eggs found in the newborn's faeces, amniotic fluid, and the mother's stool. The authors proposed an intrauterine transmission, suggesting that larvae may have crossed the placenta, entered the umbilical vein and foetal circulation, and developed into adult worms in utero. Adult female worms were also discovered in the placenta.[14]

Another case, reported by Rathi et al., described a 45-day-old infant suffering from intestinal obstruction caused by *Ascaris* infection.[15] The excretion of macerated worms precluded determining whether the worms were sexually mature or in a pre-patent stage. Possible

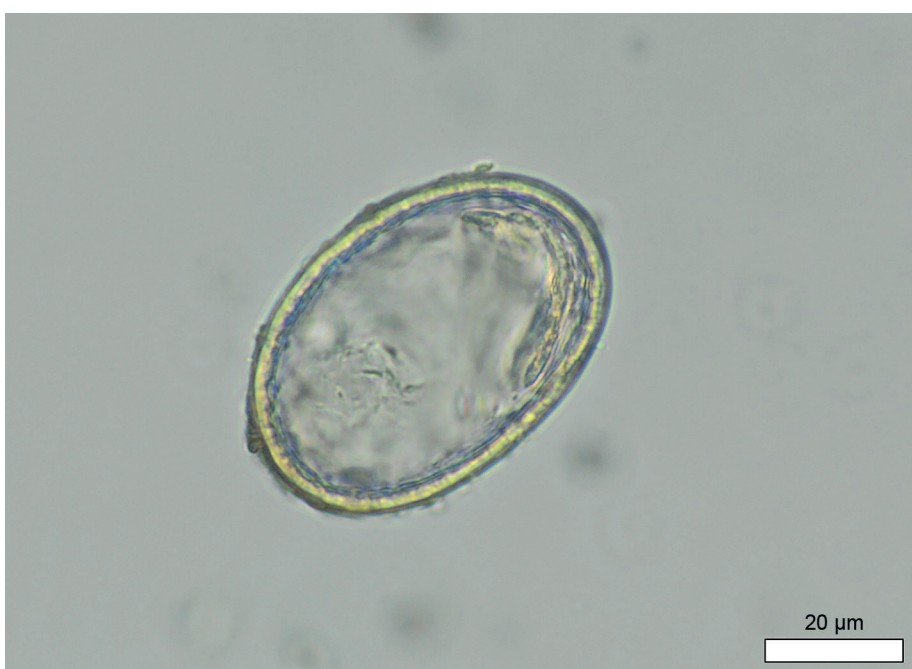

Fig. 1: Povegliano Veronese, Loc. Ortaia (Italy), burial 154. Possible female (F > M), 0-3 months. Microscopic image of one of the seven *Ascaris lumbricoides* decorticated eggs associated with the individual (measurements: 58.8 × 40.39 μm). Photo: R. López-Gijón.

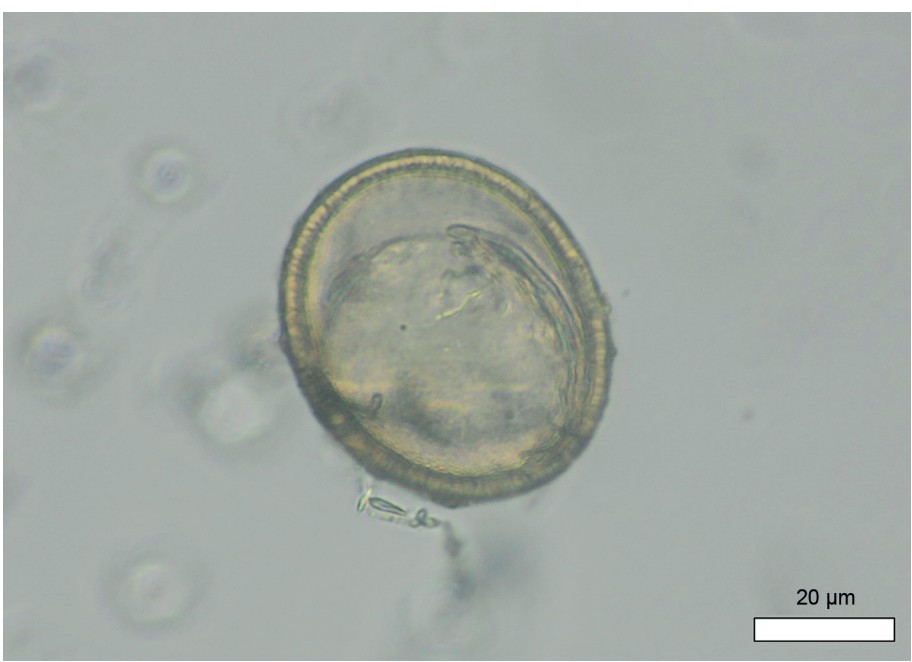

Fig. 2: Povegliano Veronese, Loc. Ortaia (Italy), burial 154. Possible female (F > M), 0-3 months. Microscopic image of another one of the seven *Ascaris lumbricoides* decorticated eggs associated with the individual (measurements: 59.08 × 48.51 μm). Photo: R. López-Gijón.

sources of infection included honey or contaminated water administered in the first days of life, though a transplacental transmission route could not be ruled out.

Further evidence of congenital transmission via the placenta is suggested in a case of neonatal ascariasis reported by Da Costa-Macedo and Rey.[16] This route of transmission is well-documented in *Toxocara canis*, another member of the Ascaridae family.[16] Neonatal *Ascaris* infections are often associated with nutritional deficiencies, including lactose digestion issues, which can severely impair weight gain in neonates.[17] For example, the neonate described by Chu et al., born at eight lunar months, weighed only 2010 g with a body length of 45 cm.[14]

In our case, precise age at death could not be determined beyond the range of 0-3 months due to inherent limitations in anthropological age estimation. Thus, the observed parasitic infection may have been congenital or acquired in the first days of life. The exact cause of death for the neonate remains undetermined. While intracranial bleeding may have been a contributing factor, we cannot exclude intestinal obstruction due to *Ascaris* infection.

The case of Povegliano contributes to the limited corpus of palaeoparasitological data available for the Iron Age in continental Europe, which is primarily represented by individual coprolites from mining sites (*e.g.*, the salt mines of Hallstatt and Hallein), intestinal samples from bog bodies (*e.g.*, the Döbritz girl and Lindow Man), and the famous early La Tène princely grave of Lavau (France) or communal samples from ditches (*e.g.*, in the Celtic settlement Basel-Gasfabrik).[18,19,20] These findings indicate that helminth infections were already widespread among Iron Age communities, occurring across diverse ecological, social, and cultural settings.

In conclusion, although *Ascaris* infections are rare in neonatal individuals, several clinical cases of neonatal ascariasis have been documented in modern contexts. However, to date, this parasite has not been identified in neonates from archaeological contexts.

In the present case, we report what may represent the oldest documented instance of ascariasis in a neonatal individual. The infection could have been acquired via intrauterine transmission or within the first weeks of life. This finding contributes to our understanding of the long-standing relationship between humans and parasitic infections, even among the most vulnerable age groups in ancient populations. A parallel study currently under review, based on additional samples from Povegliano and the nearby necropolis of Seminario Vescovile (López-Gijón et al., Unpublished data), will help better contextualise the case presented in this work.

## ACKNOWLEDGEMENTS

To the *Soprintendenza Archeologia*, *Belle Arti e Paesaggio per le Province di Verona*, *Rovigo e Vicenza* (Italy) in particular Brunella Bruno, Gianni De Zuccato, Irene Dori and Giovanna Falezza (National Archaeological Museum of Verona), for granting permission to study the material and for issuing the necessary sampling permits. We are also grateful to Giulio Squaranti and the *Associazione Balladoro* in Povegliano Veronese (Italy) for their logistical support, and to the two anonymous reviewers for their valuable comments.

## AUTHORS' CONTRIBUTION

Conceptualisation - WRT; funding acquisition - MM and AZ; excavation - DV; sampling - WRT and ZL; investigation and writing - original draft - RLG and WRT; visualisation - RLG; writing - review & editing - WRT, RLG, MM, ZL, DV and AZ. During the preparation of this work the authors used

OpenAI's ChatGPT in order to improve readability and language. After using this tool, the authors reviewed and edited the content as needed and take full responsibility for the content of the publication. No conflict of interest is stated.

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

# OPEN PEER REVIEW

Memórias do IOC thanks the anonymous reviewers for their contribution to the peer review of this work.

## FIRST REVIEW ROUND

REVIEWERS' COMMENTS

### REVIEWER #1

The abstract is suitable and the manuscript is original. However, I find crucial problems:

They postulate an unusual case: roundworm eggs in a neonatal individual, an age with a low prevalence in the literature. To support this, they show a figure with few typical features of a roundworm egg. They find only five bodies and do not describe their morphology or metric. Among the background information, at least the following should be included: Reinhard, K., Geib, P.R., Callahan, M.M., Hevly, R.H., 1992. Discovery of colon contents in a skeletonized burial: soil sampling for dietary remains. Journal of Archaeological Science 19, 697-705.

### REVIEWER #2

This study is particularly interesting, as it appears to document the earliest known case of Ascaris lumbricoides infection in a newborn, as well as the first identified occurrence of this parasite in Iron Age Italy. The findings once again underscore the importance of analysing human remains in paleoparasitology, both to enhance the existing reference base and to deepen our understanding of ancient populations. The comparison with modern parasitological data on Ascaris infections in neonates is also of great interest.

In light of these unpublished results and the new data presented in this article, its publication in Memórias do Instituto Oswaldo Cruz appears well justified. Nonetheless, I recommend the following minor revisions.

1) FINDINGS: In the 2nd sentence "Ascaris" should be italicised.

2) Introduction:

- In the 3rd paragraph "Ascaris lumbricoides" and "Trichuris trichiura" should also be italicised.

- At the end, it would be useful to include a brief description of the method employed to study the samples — ideally in a single sentence — indicating, for example, that the technique used to extract parasite markers is based on micrometric separation of elements. Moreover, the final reference cited (Ledger et al., 2021) presents results and method not only for microscopy but also for immunology.

3) Results: Are all Ascaris lumbricoides eggs observed in the samples decorticated, as illustrated in Figure 1? If so, this could be explicitly stated in the text, as it may also indicate that taphonomic processes could account the low number of eggs observed in the samples.

4) Discussion:

- 1st paragraph, 4th sentence and 2nd paragraph, 1st sentence: "Ascaris" should be italicised.

- At some point in the discussion, the authors should also address mentions of Ascaris during the Iron Age, referring, for instance, to recent syntheses in paleoparasitology for this period (Anastasiou 2015; Dufour and Le Bailly 2022). While the text rightly highlights this case as the earliest known occurrence of A. lumbricoides in a newborn, it also appears to represent the first documented instance of this taxon in humans from Iron Age Italy—a point that would be worth stated.

ANASTASIOU Evilena (2015) - « Parasites in European populations from prehistory to the industrial revolution » dans MITCHELL Piers D. (dir.) - Sanitation, Latrines and Intestinal Parasites in Past Populations. Ashgate, Farnham, p. 203-217.

DUFOUR Benjamin & LE BAILLY Matthieu (2022) - « Paléoparasitologie de l'âge du Fer dans l'ouest de l'Europe ». Revue archéologique de Picardie, n° 1-2, p. 323-329.

5) Figure caption: "Decorticated" is to be added before "Egg".

AUTHORS' RESPONSE TO THE REVIEWERS

Dear Dr. Adeilton Brandão,

We are pleased to enclose our new version of manuscript MIOC-2025-0091, which has been revised in full accordance with the suggestions and concerns of the two re viewers, to whom we are most grateful. Their kind recommendations have helped us to clarify and strengthen the paper. Our point-by point responses are listed below with an account of the corresponding modifications. Please note that two new eggs have been identified from the samples, giving a total of 7 eggs from this individual. We updated the text accordingly. To support our revised text, we further added Fig. 2.

**REVIEWER #1**

Comments: The abstract is suitable and the manuscript is original. However, I find crucial problems: They postulate an unusual case: roundworm eggs in a neonatal in dividual, an age with a low prevalence in the literature. To support this, they show a figure with few typical features of a roundworm egg. They find only five bodies and do not describe their morphology or metric.

Response: Thanks for pointing to this aspect. The recovery of intestinal helminth eggs from bur ial sediments is strongly influenced by taphonomic processes (e.g. soil acidity, humidity, tem perature), which may result in the retrieval of only a few findings. Taphonomic factors are also responsible for the appearance of archaeological decorticated eggs of Ascaris sp., the taxon most commonly represented in European archaeological contexts. In these cases indeed the charac teristic mamillated coat is missing or grossly altered (see Mitchell et al., 2013; Cunha et al., 2017; Wang et al., 2022; Marković et al., 2024). These issues are particularly frequent in Mediterranean context characterized by semi-arid climatic conditions, less favorable for parasite eggs preser vation if compared with wet environments (Anastasiou et al., 2018; Bouchet et al., 2003; Ledger et al., 2021; López-Gijón et al., 2023; Mitchell et al., 2022; Roche et al., 2019).

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

Response: We added the suggested reference.

**REVIEWER #2**

This study is particularly interesting, as it appears to document the earliest known case of Ascaris lumbricoides infection in a newborn, as well as the first identified occurrence of this parasite in Iron Age Italy. The findings once again underscore the importance of analysing human remains in paleoparasitology, both to enhance the ex isting reference base and to deepen our understanding of ancient populations. The comparison with modern parasitological data on Ascaris infections in neonates is also of great interest. In light of these unpublished results and the new data presented in this article, its publication in Memórias do Instituto Oswaldo Cruz appears well justi fied. Nonetheless, I recommend the following minor revisions.

Comments: FINDINGS: In the 2nd sentence "Ascaris" should be italicised.

Response: Done.

Comments: In the 3rd paragraph "Ascaris lumbricoides" and "Trichuris trichiura" should also be italicised.

Response: Done.

Comments: At the end, it would be useful to include a brief description of the method employed to study the samples — ideally in a single sentence — indicating, for example, that the technique used to extract parasite markers is based on micrometric separation of elements.

Response: We now expanded this section and included more details on the applied method (lines 130-133 in the tracked file = lines 92-95 in the cleaned file).

Comments: Moreover, the final reference cited (Ledger et al., 2021) presents results and method not only for microscopy but also for immunology.

Response: This is right. The sentence was therefore deleted. The reference to Ledger et al., 2021 was removed to lines 169-170 in the tracked file = lines 117-118 in the cleaned file, and rewritten as "In the neighboring Lombard burial ground of Povegliano Veronese, no para sitic eggs were detected in the 14 soil samples examined.(13= Ledger et al., 2021)".

Comments: Are all Ascaris lumbricoides eggs observed in the samples decorticated, as illustrated in Figure 1? If so, this could be explicitly stated in the text, as it may also indicate that taphonomic processes could account the low number of eggs ob served in the samples.

Response: Thanks for pointing this out. We now included this aspect to the text (lines 163-169 in the tracked file = lines 111-116 in the cleaned file). Furthermore, we added Fig. 2, another decorticated Ascaris egg, to support our text.

Comments: 1st paragraph, 4th sentence and 2nd paragraph, 1st sentence: "Ascaris" should be italicised.

Response: Done.

Comments: At some point in the discussion, the authors should also address mentions of Ascaris during the Iron Age, referring, for instance, to recent syntheses in paleo parasitology for this period (Anastasiou 2015; Dufour and Le Bailly 2022). While the text rightly highlights this case as the earliest known occurrence of A. lumbricoides in a newborn, it also appears to represent the first documented instance of this taxon in humans from Iron Age Italy — a point that would be worth stated.

ANASTASIOU Evilena (2015) - « Parasites in European populations from prehistory to the industrial revolution » dans MITCHELL Piers D. (dir.) - Sanitation, Latrines and Intestinal Parasites in Past Populations. Ashgate, Farnham, p. 203-217.

DUFOUR Benjamin & LE BAILLY Matthieu (2022) - « Paléoparasitologie de l'âge du Fer dans l'ouest de l'Europe ». Revue archéologique de Picardie, n° 1-2, p. 323-329.

Response: Thanks for this suggestion and the included references. We added a section on this aspect (lines 236-243 in the tracked file = lines 149-156 in the cleaned file). Regarding this being the first example of Ascaris for the Italian Iron Age, we preferred to be cautious about this aspect since additional samples and results will be listed in article (currently under review) focused on multiple sites in this area of Italy during the Late Iron Age.

Comments: Figure caption: "Decorticated" is to be added before "Egg".

Response: Done. Also, the measurements of both eggs (Fig. 1-2) are given.

I would be very happy, also on behalf of my co-authors, if the revised manuscript would now be accepted.

With kind regards

Prof. Dr. Wolf-Rüdiger Teegen

---

**SECOND REVIEW ROUND**

**REVIEWERS' COMMENTS**

---

**REVIEWER #1**

The original research presented in this article remains as engaging as ever. In my view, the authors have addressed all the reviewers' comments and questions in a satisfactory and appropriate manner.

Having carefully reviewed the authors' responses and the revisions made to the manuscript, I find no further points requiring modification or addition. In my opinion, the article is ready for publication in its current form.

