## [Reviewer Report · FIRST REVIEW ROUND - REVIEWERS COMMENTS]

## Reviewer #1

The abstract is suitable and the manuscript is original. However, I find crucial problems:

They postulate an unusual case: roundworm eggs in a neonatal individual, an age with a low prevalence in the literature. To support this, they show a figure with few typical features of a roundworm egg. They find only five bodies and do not describe their morphology or metric. Among the background information, at least the following should be included: Reinhard, K., Geib, P.R., Callahan, M.M., Hevly, R.H., 1992. Discovery of colon contents in a skeletonized burial: soil sampling for dietary remains. Journal of Archaeological Science 19, 697-705.

## Reviewer #2

This study is particularly interesting, as it appears to document the earliest known case of Ascaris lumbricoides infection in a newborn, as well as the first identified occurrence of this parasite in Iron Age Italy. The findings once again underscore the importance of analysing human remains in paleoparasitology, both to enhance the existing reference base and to deepen our understanding of ancient populations. The comparison with modern parasitological data on Ascaris infections in neonates is also of great interest.

In light of these unpublished results and the new data presented in this article, its publication in Memórias do Instituto Oswaldo Cruz appears well justified. Nonetheless, I recommend the following minor revisions.

1) FINDINGS: In the 2nd sentence “Ascaris” should be italicised.

2) Introduction:

- In the 3rd paragraph “Ascaris lumbricoides” and “Trichuris trichiura” should also be italicised.

- At the end, it would be useful to include a brief description of the method employed to study the samples — ideally in a single sentence — indicating, for example, that the technique used to extract parasite markers is based on micrometric separation of elements. Moreover, the final reference cited (Ledger et al., 2021) presents results and method not only for microscopy but also for immunology.

3) Results: Are all Ascaris lumbricoides eggs observed in the samples decorticated, as illustrated in Figure 1? If so, this could be explicitly stated in the text, as it may also indicate that taphonomic processes could account the low number of eggs observed in the samples.

4) Discussion:

- 1st paragraph, 4th sentence and 2nd paragraph, 1st sentence: “Ascaris” should be italicised.

- At some point in the discussion, the authors should also address mentions of Ascaris during the Iron Age, referring, for instance, to recent syntheses in paleoparasitology for this period (Anastasiou 2015; Dufour and Le Bailly 2022). While the text rightly highlights this case as the earliest known occurrence of A. lumbricoides in a newborn, it also appears to represent the first documented instance of this taxon in humans from Iron Age Italy—a point that would be worth stated.

ANASTASIOU Evilena (2015) - « Parasites in European populations from prehistory to the industrial revolution » dans MITCHELL Piers D. (dir.) - Sanitation, Latrines and Intestinal Parasites in Past Populations. Ashgate, Farnham, p. 203-217.

DUFOUR Benjamin e LE BAILLY Matthieu (2022) - Paléoparasitologie de l’âge du Fer dans l’ouest de l’Europe. Revue archéologique de Picardie, n° 1-2, p. 323-329.

5) Figure caption: “Decorticated” is to be added before “Egg”.

---

## [Author Response · AUTHORS RESPONSE TO REVIEWERS]

## Dear Dr. Adeilton Brandão,

We are pleased to enclose our new version of manuscript MIOC-2025-0091, which has been revised in full accordance with the suggestions and concerns of the two reviewers, to whom we are most grateful. Their kind recommendations have helped us to clarify and strengthen the paper. Our point-by point responses are listed below with an account of the corresponding modifications. Please note that two new eggs have been identified from the samples, giving a total of 7 eggs from this individual. We updated the text accordingly. To support our revised text, we further added Fig. 2.

---

## [Reviewer Report · REVIEWER 1]

## Reviewer #1

Comments: The abstract is suitable and the manuscript is original. However, I find crucial problems: They postulate an unusual case: roundworm eggs in a neonatal individual, an age with a low prevalence in the literature. To support this, they show a figure with few typical features of a roundworm egg. They find only five bodies and do not describe their morphology or metric.

Response: Thanks for pointing to this aspect. The recovery of intestinal helminth eggs from burial sediments is strongly influenced by taphonomic processes (e.g. soil acidity, humidity, temperature), which may result in the retrieval of only a few findings. Taphonomic factors are also responsible for the appearance of archaeological decorticated eggs of Ascaris sp., the taxon most commonly represented in European archaeological contexts. In these cases indeed the characteristic mamillated coat is missing or grossly altered (see Mitchell et al., 2013; Cunha et al., 2017; Wang et al., 2022; Marković et al., 2024). These issues are particularly frequent in Mediterranean context characterized by semi-arid climatic conditions, less favorable for parasite eggs preservation if compared with wet environments (Anastasiou et al., 2018; Bouchet et al., 2003; Ledger et al., 2021; López-Gijón et al., 2023; Mitchell et al., 2022; Roche et al., 2019).

References:

Anastasiou E, Papathanasiou A, Schepartz LA, Mitchell PD. Infectious disease in the ancient Aegean: Intestinal parasitic worms in the Neolithic to Roman Period inhabitants of Kea, Greece. J. Archaeol. Sci. Rep. 2018; 17, 860-864. https://doi.org/10.1016/j.jasrep.2017.11.006

Bouchet F, Harter S, Le Bailly M. The state of the art of paleoparasitological research in the Old World. Mem. Inst. Oswaldo Cruz. 2003; 98, 95–101. https://doi.org/ 10.1590/S0074 02762003000900015

Cunha D, Santos AL, Matias A, Sianto L. A novel approach: combining dental enamel hypoplasia and paleoparasitological analysis in medieval Islamic individuals buried in Santarém (Portugal). Antropol. Port. 2017; 34, 113–135. https://doi.org/ 10.14195/2182-7982_34_6

Ledger ML, Micarelli I, Ward D, Prowse TL, Carroll M, Killgrove K, Rice C, Franconi T, Tafuri MA, Manzi G, Mitchell PD. Gastrointestinal infection in Italy during the Roman Imperial and Longobard periods: A paleoparasitological analysis of sediment from skeletal remains and sewer drains. Int. J. Paleopathol. 2021; 33, 61–71. https://doi.org/10.1016/j.ijpp.2021.03.001

López-Gijón R, Duras S, Botella-López MC, Sentí-Ribes MA, Dufour B, Le Bailly M. Evidencia paleoparasitológica de Ascaris lumbricoides en restos esqueletizados de época romana Dianium (Alicante, https://doi.org/10.21630/maa.2022.73.10 España). Munibe. 2022; 73, 181–190.

Marković N., Savić AR, Mitić A, Mitchell PD. Palaeoparasitological evidence for a possible sanitary stone vessel from the Roman city of Viminacium, Serbia. J. Archaeol. Sci. Rep. 2024; 57, 104671. https://doi.org/10.1016/j.jasrep.2024.104671

Mitchell PD, Yeh HY, Appleby J, Buckley R. The intestinal parasites of King Richard III. Lancet. 2013; 382(9895), 888. http://dx.doi.org/10.1016/S0140-6736(13)61757-2

Mitchell PD, Anastasiou E, Whelton HL, Bull ID, Pearson MP, Shillito LM. Intestinal parasites in the Neolithic population who built Stonehenge (Durrington Walls, 2500 BCE). Parasitology. 2022; 149, 1027–1033. https://doi.org/ 10.1017/S0031182022000476

Roche K, Pacciani E, Bianucci R, Le Bailly M. Assessing the parasitic burden in a Late Antique Florentine emergency burial site. Korean J. Parasitol. 2019; 57, 587–593 https://doi.org/10.3347%2Fkjp.2019.57.6.587

Wang T, Cessford C, Dittmar JM, Inskip S, Jones PM, Mitchell PD. Intestinal parasite infection in the Augustinian friars and general population of medieval Cambridge, UK. Int. J. Paleopathol. 2022; 39, 115-121. https://doi.org/10.1016/j.ijpp.2022.06.001

Comments: They find only five bodies and do not describe their morphology or metric.

Response: We now expanded the results section and are more explicit about the criteria underlying the taxonomic attribution of these finds (lines 163-169 in the tracked file = lines 111-116 in the cleaned file).

Comments: Among the background information, at least the following should be included: Reinhard, K., Geib, P.R., Callahan, M.M., Hevly, R.H., 1992. Discovery of colon contents in a skeletonized burial: soil sampling for dietary remains. Journal of Archaeological Science 19, 697-705.

Response: We added the suggested reference.

## Reviewer #2

This study is particularly interesting, as it appears to document the earliest known case of Ascaris lumbricoides infection in a newborn, as well as the first identified occurrence of this parasite in Iron Age Italy. The findings once again underscore the importance of analysing human remains in paleoparasitology, both to enhance the existing reference base and to deepen our understanding of ancient populations. The comparison with modern parasitological data on Ascaris infections in neonates is also of great interest. In light of these unpublished results and the new data presented in this article, its publication in Memórias do Instituto Oswaldo Cruz appears well justified. Nonetheless, I recommend the following minor revisions.

Comments: FINDINGS: In the 2nd sentence “Ascaris” should be italicised.

Response: Done.

Comments: In the 3rd paragraph *Ascaris lumbricoides* and *Trichuris trichiura* should also be italicised.

Response: Done.

Comments: At the end, it would be useful to include a brief description of the method employed to study the samples — ideally in a single sentence — indicating, for example, that the technique used to extract parasite markers is based on micrometric separation of elements.

Response: We now expanded this section and included more details on the applied method (lines 130-133 in the tracked file = lines 92-95 in the cleaned file).

Comments: Moreover, the final reference cited (Ledger et al., 2021) presents results and method not only for microscopy but also for immunology.

Response: This is right. The sentence was therefore deleted. The reference to Ledger et al., 2021 was removed to lines 169-170 in the tracked file = lines 117-118 in the cleaned file, and rewritten as “In the neighboring Lombard burial ground of Povegliano Veronese, no parasitic eggs were detected in the 14 soil samples examined.(13= Ledger et al., 2021)“.

Comments: Are all Ascaris lumbricoides eggs observed in the samples decorticated, as illustrated in Figure 1? If so, this could be explicitly stated in the text, as it may also indicate that taphonomic processes could account the low number of eggs observed in the samples.

Response: Thanks for pointing this out. We now included this aspect to the text (lines 163-169 in the tracked file = lines 111-116 in the cleaned file). Furthermore, we added Fig. 2, another decorticated Ascaris egg, to support our text.

Comments: 1st paragraph, 4th sentence and 2nd paragraph, 1st sentence: “Ascaris” should be italicised.

Response: Done.

Comments: At some point in the discussion, the authors should also address mentions of Ascaris during the Iron Age, referring, for instance, to recent syntheses in paleoparasitology for this period (Anastasiou 2015; Dufour and Le Bailly 2022). While the text rightly highlights this case as the earliest known occurrence of A. lumbricoides in a newborn, it also appears to represent the first documented instance of this taxon in humans from Iron Age Italy — a point that would be worth stated.

ANASTASIOU Evilena (2015) - « Parasites in European populations from prehistory to the industrial revolution » dans MITCHELL Piers D. (dir.) - Sanitation, Latrines and Intestinal Parasites in Past Populations. Ashgate, Farnham, p. 203-217.

DUFOUR Benjamin e LE BAILLY Matthieu (2022) - Paléoparasitologie de l’âge du Fer dans l’ouest de l’Europe. Revue archéologique de Picardie, n° 1-2, p. 323-329.

Response: Thanks for this suggestion and the included references. We added a section on this aspect (lines 236-243 in the tracked file = lines 149-156 in the cleaned file). Regarding this being the first example of Ascaris for the Italian Iron Age, we preferred to be cautious about this aspect since additional samples and results will be listed in article (currently under review) focused on multiple sites in this area of Italy during the Late Iron Age.

Comments: Figure caption: “Decorticated” is to be added before “Egg”.

Response: Done. Also, the measurements of both eggs (Fig. 1-2) are given.

I would be very happy, also on behalf of my co-authors, if the revised manuscript would now be accepted.

With kind regards

Prof. Dr. Wolf-Rüdiger Teegen

---

## [Reviewer Report · REVIEWERS COMMENTS]

## Reviewer #1

The original research presented in this article remains as engaging as ever. In my view, the authors have addressed all the reviewers’ comments and questions in a satisfactory and appropriate manner.

Having carefully reviewed the authors’ responses and the revisions made to the manuscript, I find no further points requiring modification or addition. In my opinion, the article is ready for publication in its current form.